# Self-Management in Stroke Survivors: Development and Implementation of the Look after Yourself (LAY) Intervention

**DOI:** 10.3390/ijerph18115925

**Published:** 2021-05-31

**Authors:** Stefania Fugazzaro, Monica Denti, Monia Allisen Accogli, Stefania Costi, Donatella Pagliacci, Simona Calugi, Enrica Cavalli, Mariangela Taricco, Roberta Bardelli

**Affiliations:** 1Physical Medicine and Rehabilitation Unit, Azienda Unità Sanitaria Locale-IRCCS di Reggio Emilia, Viale Risorgimento n°80, 42123 Reggio Emilia, Italy; stefania.fugazzaro@ausl.re.it (S.F.); moniaallisen.accogli@ausl.re.it (M.A.A.); 2Scientific Directorate, Azienda Unità Sanitaria Locale-IRCCS di Reggio Emilia, Viale Umberto I n°50, 42123 Reggio Emilia, Italy; stefania.costi@unimore.it; 3Department of Surgery, Medicine, Dentistry and Morphological Sciences, University of Modena and Reggio Emilia, Via del Pozzo n°74, 41100 Modena, Italy; 4Department of Community Health Care, Azienda Unità Sanitaria Locale Toscana Nord-Ovest, Via A. Cocchi n°7/9, 56124 Pisa, Italy; donapaglia@gmail.com; 5Department of Biomedical and Neuromotor Sciences, University of Bologna, Via Ugo Foscolo, n°7, 40123 Bologna, Italy; si.calugi@gmail.com; 6Physical Medicine and Rehabilitation Unit, Azienda Ospedaliero-Universitaria Policlinico S.Orsola-Malpighi, Via Albertoni n°15, 40138 Bologna, Italy; enricacavalli@hotmail.com (E.C.); mariangelataricco@gmail.com (M.T.); 7Physical Medicine and Rehabilitation Unit, IRCCS Istituto Ortopedico Rizzoli, via Giulio Cesare Pupilli n°1, 40136 Bologna, Italy; roberta.bardelli@ior.it

**Keywords:** stroke rehabilitation, self-management, self-efficacy, chronic disease, patient education

## Abstract

Objective: Self-management is recommended in stroke rehabilitation. This report aims to describe timing, contents, and setting of delivery of a patient-centered, self-management program for stroke survivors in their early hospital rehabilitation phase: the Look After Yourself (LAY) intervention. Methods: After extensive literature search, the LAY intervention was developed by integrating the Chronic Disease Self-Management Program, based on the self-efficacy construct of social cognitive theory, with evidence-based key elements and input from stroke survivors. Results: the LAY intervention aims to implement self-management skills in stroke survivors, enabling them to be active in goal setting and problem solving using action plans and to facilitate the critical transition from hospital to community. It includes both group sessions to facilitate sharing of experiences, social comparison, vicarious learning, and increase motivation and one-to-one sessions focused on setting feasible action plans and on teaching personalized strategies to prevent falls. Standardization is ensured by manuals for facilitators and patients. Conclusion: The LAY intervention is the first Italian program to support early self-management in stroke rehabilitation; it has been experimented and its efficacy proven in improving self-efficacy, mental health, and activities of daily living, and detailed results have been published. The LAY intervention is described according to the TIDieR checklist.

## 1. Introduction

Stroke is the second most common cause of death and a leading cause of adult physical disability, affecting 17 million people worldwide each year. Its incidence and prevalence are increasing, with burden on both stroke survivors’ quality of life and on health systems [1,2]. Current stroke care management supports early discharge from hospital when rehabilitation is still under way [3,4,5]. Nevertheless, studies have underlined that stroke survivors and their caregivers often feel unprepared to face the transition from hospital to community [6,7]

International stroke guidelines have recommended that “All patients should be offered training in self-management skills, including active problem solving and individual goal setting” [8,9]. Thus, in recent years, self-management has become part of the stroke care pathway, since it could support individuals facing the long-term consequences of stroke [10], and thus it could facilitate interventions related to transitional care [11].

Over the past five years, our research group has conducted a project funded by the Emilia Romagna Region: Look After Yourself (LAY)—an educational intervention for stroke patients to improve self-management and fostering transition from hospital to community. The first phase of the project was dedicated to the development of this educational intervention and to the training of healthcare professionals, while the second phase consisted in a quasi-experimental study. This was conducted on a sample of 185 post-acute stroke patients recruited from three different rehabilitation centers: two implemented the experimental intervention, and one acted as control (ISRCTN75290225).

Given the nature of the intervention (educational), which led to a global reorganization in the two experimental centers, it would not be appropriate to set up a randomized controlled trial. This is because, in those two experimental centers, all patients, even those who did not participate in the research, were exposed to the learning of new skills they acquired from the trained professionals, as usually happens when complex and long-term educational interventions are promoted.

The results of this study indicate that the LAY intervention improves self-efficacy, mental health, and activity of daily living. There were no side effects, and overall user/professional satisfaction was good. There was therefore evidence to support the implementation of self-management programs in stroke survivors. The design and results of this quasi-experimental study have been described in a recently published paper [12].

The aim of this paper is to describe in detail phase one of the LAY project, explaining how the LAY intervention has been built, developed, and implemented in the context of three Italian rehabilitation centers. The purpose is to spread the knowledge acquired in order to encourage the use of self-management programs and their adaptation to different contexts.

## 2. Intervention Description

The TIDieR checklist [13] was used to structure the detailed description of the LAY intervention. 

### 2.1. Item 1. Brief Name

The LAY intervention is a self-management program that aims to improve self-efficacy and foster transition from hospital to community of stroke survivors in post-acute phase. It is an Italian adaptation from the Chronic Disease Self-Management Program (CDSMP), which is a standardized, evidence-based intervention for the comprehensive self-management of chronic conditions. It is based on the self-efficacy theory [14,15] and was developed at the Stanford Patient Education Research Center of Stanford University.

### 2.2. Item 2. Why: Description of the Rational and Theory Essential to the Intervention

The LAY intervention was developed by a multi-professional research group composed of physicians, physiotherapists, nurses and psychologists. The development of the LAY intervention consisted of three steps: -Identification of the evidence base for the LAY intervention.-Set-up of the LAY intervention in terms of contents, duration, and delivery.-Test the relevance of contents and the feasibility of sessions and the fine-tuning review process.

In the first step, an extensive literature search for the best evidence on self-management programs targeting stroke patients and their needs in the early stage was conducted. Then, an in-depth analysis of the literature retrieved was made to identify the key elements of the existing programs in order to inform the LAY intervention.

The literature search yielded a systematic review by Lennon [16], published in 2013, which extensively examined the evidence of self-management programs specific for stroke survivors. It included nine randomized, controlled trials highlighting the potential importance of self-management but also underlining relevant differences in timing, content, and mode of delivering the self-management interventions in this population. Moreover, a further randomized clinical trial aimed at verifying the feasibility of a self-management program for stroke survivors in the community setting was retrieved [17]. Table 1 shows the main characteristics of the trials that informed the LAY intervention, whose quality has been assessed by way of a critical appraisal tool developed by The American Academy for Cerebral Palsy and Developmental Medicine [18].

Although statistically significant findings in favour of the self-management programs were found in most of the studies examined and reported in Table 1, some criticisms were highlighted, such as small study samples (<100 participants) [17,23] and, in general, a poor description of the interventions applied.

To be noted, four trials experimented with self-management interventions based on principles of self-efficacy [17,23,24,25], and three other trials offered self-management interventions based on the theoretical model of the Chronic Disease Self-Management Program (CDSMP) [21,22,26]. The CDSMP is one of the most internationally widespread interventions to support self-management [28]; it consists of six weekly group sessions and is founded on three main assumptions: (1) individuals with different chronic diseases share similar self-management problems and disease-related tasks; (2) individuals can learn to take responsibility for the day-to-day management of their disease; (3) individuals confident and knowledgeable in practicing self-management will improve their health status. Workshops can be led by peer-leaders or by healthcare professionals and are directed towards patients and their caregivers. The CDSMP, originally developed to promote self-management among individuals with a variety of chronic conditions [29], has also been applied to diabetes [30] and cancer [31], and it is currently being broadly disseminated across various countries [32].

Among the self-management programs based on the CDSMP, the Stroke Self-Management Program (SSMP) implemented by Damush and collaborators [22] is an intervention addressing the very early phase needs of patients with stroke. The SSMP, which was developed by the National Stroke Foundation, proved to be feasible and beneficial in veterans enrolled in the program within one month post-stroke; it was delivered in six sessions, both one-to-one (three sessions) and by telephone (three sessions), over a three-month period. The feasibility of this program is to be highlighted, as stroke survivors experience sudden, considerable, and frequently long-lasting disability that often spreads to the cognitive or communication areas. Thus, full participation in self-management programs can be challenging, especially in the early rehabilitation phase. Thus, self-management programs specifically adapted to stroke survivors’ needs are now recommended, and the evidence in this area of research is growing [33,34]. Furthermore, most of the structured self-management programs directed towards stroke survivors have been developed in English-speaking countries [34]; until today, no such program has ever existed in Italy. 

### 2.3. Item 3. What: Description of Materials

The LAY intervention was set up to match stroke survivors’ needs and context elements, taking into account the clinically relevant issues typical of the early rehabilitation phase and features of the in-hospital rehabilitation setting and organization. 

The LAY intervention is an adaptation of the CDSMP to stroke patients and their caregivers, and its contents and overview are described in Table 2. The research group asked and obtained a research license from the Stanford University before drawing inspiration from the CDSMP contents.

The main tool of the LAY intervention is the action plan, as in the CDSMP. The action plan helps the patient to set and achieve his/her personal goals, which have to be specific, measurable, achievable, realistic, and need to be addressed to a relevant action that the individual is motivated to carry out in the short-term (SMART) [35]. In order to do this, patients must answer a number of questions establishing how, when, and how often they plan to perform the action, anticipating any helpful aid they might need as well as any possible causes of failure (in order to prevent it) and/or risks to their safety. Lastly, patients must rate their level of confidence about the success of the action planned on a ten-point scale; if the score is lower than seven, the action plan has to be modified in order to get higher levels of confidence [36].

In order to standardize and to give the possibility to replicate the intervention, two manuals were developed: one for program leaders (physicians, physiotherapists, and nurses), which is a guide to conduct the group and individual sessions and one for participants (stroke survivors and their caregivers), which describes the topics addressed during the sessions. 

The program leader’s manual was informed by both the original 2012 version and the Italian version of the CDSMP Leader’s Manual, integrated with the standardized manual of SSMP [22] and recommendations provided by Teresa Damush. 

The participants’ manual was informed by different sources, such as the Living a Healthy Life with Chronic Conditions [36] and the National Stroke Association materials [37]. This manual contains the action plan templates and an activity diary that the patient is meant to fill in individually. It also includes information on resources available in the community to support the social reintegration of individuals after hospital discharge.

Both manuals are available in Italian version, on request to the authors.

### 2.4. Item 4. What: Procedures Followed

Stroke survivors and their caregivers were invited to participate in the LAY intervention within the first 1–2 weeks after discharge from the stroke unit, during the in-hospital rehabilitation phase. The program consists of six weekly group sessions for patients and their caregivers plus two one-to-one sessions (Figure 1). Participants who were discharged from the hospital before program completion attended the group sessions as outpatients.

The LAY intervention was delivered following the schedule reported below:-Week 1: group session 0.-Week 2: individual session 1 and group session 1.-Week 3: group session 2.-Week 4: group session 3.-Week 5: individual session 2 and group session 4.-Week 6: group session 5.

The group sessions shared a common schedule; session 0 differed from the other in the fact that action plans were not discussed, because this tool was presented to the patient in the first individual session (Table 2).

### 2.5. Item 5. Providers

The group sessions were conducted by two healthcare professionals included in the research group composed by 2 physicians, 2 physiotherapists, and 2 nurses (program leaders). The professionals were present in turn week after week. 

Individual session 1 was conducted by one healthcare professional in turn, while individual session 2, which focused on fall prevention and balance exercises, was always conducted by a physiotherapist.

Two levels of training were offered in the two hospitals where the experimental intervention was carried out. The first level was addressed to all the healthcare professionals working in the rehabilitation wards and focused on key self-management elements, communication skills, and practice in collaborative goal setting with stroke survivors and their caregivers. The second level of training was addressed to the program leaders of the research group. It focused on small group management, deepening of contents of the basic training, and practice with focus group in leading group sessions. 

The first and the second level of training were led by a psychologist and a national expert certified trainer in CDSMP (Master Trainer in 2009; T Trainer in 2014) (DP), who also contributed to the Italian network supporting self-management in chronic conditions and participated in the two-year project of Diabetes Self-Management Program, funded by the Italian Ministry of Health.

Furthermore, all the physiotherapists working in the two hospitals where the experimental intervention was carried out received specific training in patient education on accidental falls prevention, which was delivered by the two physiotherapists of the research group.

### 2.6. Item 6. LAY Intervention Delivery

#### 2.6.1. Group Sessions

The group sessions lasted 1–1.5 h, were led by two program leaders, and were held in the early afternoon. In order to foster the participation of stroke survivors, who may experience attention deficits in their post-acute, early-rehabilitation phase, groups were composed of a maximum of ten participants (smaller than in the CDSMP).

Group session 0 was the only one opened to all stroke patients hospitalized for rehabilitation, not just to the participants of the LAY Project.

Sessions had to be attended in sequence because the topics addressed were connected to the changing condition of patients during the recovery process.

#### 2.6.2. One-to-One Sessions

The two one-to-one sessions lasted about 20–30 min and were planned in the morning during the hospitalization period. The first session took place between group sessions 0 and 1, and it was led by a program leader who supported participant to make the first action plan. The goal of the program was to empower the participant to establish his/her own achievable action plans focused on relevant goals. The following action plans could be made by participants themselves or with the support of their caregiver or a trained healthcare professional. Participants were taught to plan their action plan after each group session and to share their results at the following one. Action plans could be made in hospital, while patients were hospitalized, or at home afterwards. 

The second one-to-one session was led by a trained physiotherapist; it focused on accidental fall prevention and was always planned before hospital discharge and before group session four, when possible.

### 2.7. Item 7. Where: Type of Location 

Both group and individual sessions occurred in hospital in dedicated locations of the rehabilitation wards involved.

Group sessions took place in a large room, such as the rehabilitation gym or meeting room depending on number of participants. The rooms were equipped with chairs arranged in semicircle, one personal computer, one projector, and one flipchart with markers.

### 2.8. Item 8. When and How Much 

The timing of intervention, schedule, and delivery were already described in Items 4 and 6. A total of 32 group sessions and 112 individual sessions were done between June 2015 and March 2017.

### 2.9. Item 9. Tailoring 

The LAY intervention is an adaptation of the CDSMP to post-acute stroke patients: topics, timing, format, and strategies to lead group and individual sessions were defined to match the specific needs of stroke survivors and their changes during the recovery process. 

The adaptation consisted in integrating the CDSMP with: (a) evidence-based, key elements of self-management programs experimented in stroke survivors; (b) inputs from 6 focus groups made with 3 individuals recently discharged from rehabilitation after a stroke and their caregivers, who contributed suggestions regarding the relevance of the LAY contents and the feasibility of the sessions; (c) expertise in stroke rehabilitation of clinician included in the research group; (d) inputs from Teresa Damush, SSMP developer, who provided the research group with an overview of her program and useful recommendations on self-management delivery strategies for stroke survivors.

The LAY intervention adaptation took into account the peculiarities of patients affected by stroke: -During the first few weeks after the event, individuals need time to understand what has happened, so the program focuses on the development of coping and adaptation strategies from the very first rehabilitation phase.-Duration of each group session was reduced compared to CDSMP, because during the in-hospital phase, stroke survivors frequently require long periods of rest, as they experience a lack of energy, defined as post-stroke fatigue, which negatively impacts participation in activities [38,39].-Furthermore, in the post-acute phase, stroke survivors may experience reduced attention span, memory capacities, and communication deficits; for these reasons, the CDSMP contents were simplified and individual sessions were introduced.-Both the individual sessions and the action plan guarantee the tailoring of the intervention to each patient because the individual sessions targeted at accidental fall prevention explored the patient’s specific performance and context, and because the action plan trained the individuals to identify their own significant goals and to solve their specific problems.

At the end of this adaptation course, the whole research group participated in the fine-tuning review process, and a final consensus on the LAY intervention was reached.

### 2.10. Item 10. Modifications during the Course of the Study

No change of the planned intervention was made during the course of the study.

### 2.11. Item 11. How Well (Planned) 

Patients’ adherence to group and individual sessions was assessed by the research group through the detection of presence at the session theoretically planned for each patient. In case of absence, participants were contacted and invited to the following session. These data have been previously reported [12].

### 2.12. Item 12. How Well (Actual) 

The adherence was considered good if patients attended at least 6 over the 8 total sessions provided (both group and individual) and has been previously reported [12].

## 3. Discussion

The LAY intervention is a structured self-management program directed towards stroke survivors and includes the five self-management skills described by Lorig and Holman: problem solving, decision making, appropriate resource utilization, partnership with healthcare professionals, and implementation of actions necessary to manage health issues autonomously [40].

The mediator between self-management skills and the proper self-management behaviours is self-efficacy, that is, the individual’s perception of one’s own confidence in the ability to carry out specific, health-promoting behaviours [15]. Table 3 describes the LAY key elements addressing self-efficacy and self-management.

The LAY intervention, an adaptation of the CDSMP, targets stroke survivors and is delivered from the early, post-acute phase of rehabilitation.

The main adaptations made to meet stroke survivors’ needs consists in the integration of stroke-specific themes with issues common to other chronic diseases, the delivery of simplified contents in quite concise manner to a limited number of participants, and the introduction of one-to-one sessions to personalize the intervention, matching the clinical features of each participant, and also through the action plan implementation.

These adaptations were made to suit the LAY intervention to the early post-acute phase of stroke, when individuals are still facing subacute problems, and may present a limited attention span, communication impairment, and fatigue. This requires short group sessions, plain language, and flexibility. Compared to large groups, small groups allow program leaders to pay more careful attention to each participant and to allow for in-depth interaction also with participants with physical and emotional frailty.

The action plan, that is the core of the LAY intervention, is a simple tool to enhance problem-solving and goal-setting skills, and it can be used both in hospital and in the community. The LAY intervention was set up to be delivered in the first few weeks after stroke to facilitate the critical transition from hospital to community, providing individuals with skills to control their global health condition by self-managing disease symptoms and risk factors, emotional consequences of illness, and role management (how to maintain a previous social and family role or how to create a new one). 

As stroke can result in permanent disability which can lead to social isolation, the LAY intervention incorporates information regarding resources locally available offered by community or by patient associations (transportation, events, sports facilities, etc.). The repetition of such an intervention requires consideration of the specific context and adaptation of only the chapter that describes community resources in order to fix the information to the local situation. 

A particular focus is on training required to deliver the intervention: our research group was supported by a psychologist and a national expert certified trainer in CDSMP, who planned two levels of training: the first addressed to all the health professionals of the rehabilitation wards involved in the project and the second designed for LAY sessions leaders.

A lesson we learned is that a great cultural change is needed for healthcare professionals to let patients take health in their hands; not all healthcare professionals are ready to share responsibility and power with people. Stroke rehabilitation teams still work mainly on therapy-led, multidisciplinary goal setting: physiotherapists and nurses identify the problems, define the goals, assess whether they have been achieved, and decide how the patients should progress [41]. This makes clinicians only partially meet the recommendations in national clinical guidelines.

Collaborative goal setting is a central element in rehabilitation, but it requires skills and training. This is a major challenge for future if we trust in patients’ engagement.

### 3.1. Limitations of the Intervention

Despite having simplified the original program, this kind of intervention might still be difficult for stroke patients with severe aphasia or cognitive impairment. This subgroup of patients is often excluded from research studies on innovative approaches because of comprehension and/or communication barriers. Therefore, as for other similar interventions, the generalizability of the LAY intervention cannot be immediately extended to the whole population with stroke.

Because the LAY intervention is conducted in the hospital setting, this facilitates participation of frail patients in the very early stages of their recovery after stroke, even though a certain degree of flexibility is always recommended to adapt to the unstable clinical condition and the need for intensive care. However, patient participation might be hindered after hospital discharge if residual limitation in autonomy and mobility could, by itself, prevent participation in the sessions.

Moreover, the LAY intervention was designed to be almost completely delivered by healthcare professionals, with small peer-leader representation. The research group involved patient associations in the development of the LAY intervention, but their active role in the provision of the intervention was limited to the last group session, when community services were presented. Introducing a peer-educator within the program might provide ongoing self-management support, improve the patients’ level of self-efficacy, and assist patients in dealing with the emotional components associated with their chronic condition [42].

### 3.2. Strengths of the Intervention

A strength of the LAY intervention is its mixed format, which includes both one-to-one and group sessions. One-to-one sessions allow participants to learn how to use an action plan to plan actions focusing on health goals. Using the action plan to set clinically relevant and realistic goals is reinforced during group sessions.

Furthermore, the second one-to-one session, on fall prevention, is led by the physiotherapist and actually follows the patient during clinical rehabilitation; this makes it possible to personalize the information and to teach appropriate balance and resistance exercises to the individual patient.

Concerning group sessions, their repetitive structure helps patients to reinforce the main principles and topics of the program. Group sessions have an important role for peer support, for example, in discussing successful/unsuccessful action plans within the group and in sharing ideas and advice for the next action to be planned. 

As highlighted by a recent systematic review [43], increasing knowledge, effective collaboration and/or communication, accessing resources, goal setting and problem solving, and peer support are common key features of self-management interventions and are also present in the LAY intervention. In particular, peer support among stroke survivors facilitates the sharing of experiences, social comparison, vicarious learning, and it increases motivation. Vicarious learning, in turn, influences self-perception related to one’s own ability to self-manage stroke outcomes [43].

Another strength of the LAY intervention is the early timing of delivery after stroke; the program matches the great need for information that stroke survivors and their caregivers report since the very first weeks after the stroke. A recent systematic meta-review [44] confirms that self-management interventions in stroke survivors, delivered soon after the event (<1 year), reduce patients’ dependence/institutional care or death, are beneficial to daily living activities, and seem to facilitate reintegration in the community. A positive aspect of the LAY intervention is the inclusion of caregivers in group sessions as facilitators of patient self-management behaviors. We consider of great value the presence of caregivers for their role in providing the family with assistance and support, as stroke is a life-changing event that often causes long-term disability.

Finally, the LAY intervention has been completely described using the TIDieR guide [13], which allows for reliable implementation and potential replication in similar contexts, and adaptation to different ones.

## 4. Conclusions

The LAY intervention is the first structured Italian program to support self-management for stroke survivors in their early rehabilitation phase. Since self-management is strongly related to recovery after stroke [45] the LAY intervention could support the critical transition from hospital to community in the stroke survivors’ care pathway.

The results of the LAY project, including patients’ adherence to the program, changes in self-efficacy, modification in activity of daily life, quality of life, resource utilization, and other outcome measures, support the implementation of structured self-management interventions in the rehabilitation process of stroke survivors [12].

In this line of research, more insight is needed to explore the barriers to and opportunities for delivering self-management interventions in post-acute stroke settings. Furthermore, investigations should assess the feasibility and efficacy of self-management interventions across secondary, primary, and community settings.

## Figures and Tables

**Figure 1 ijerph-18-05925-f001:**
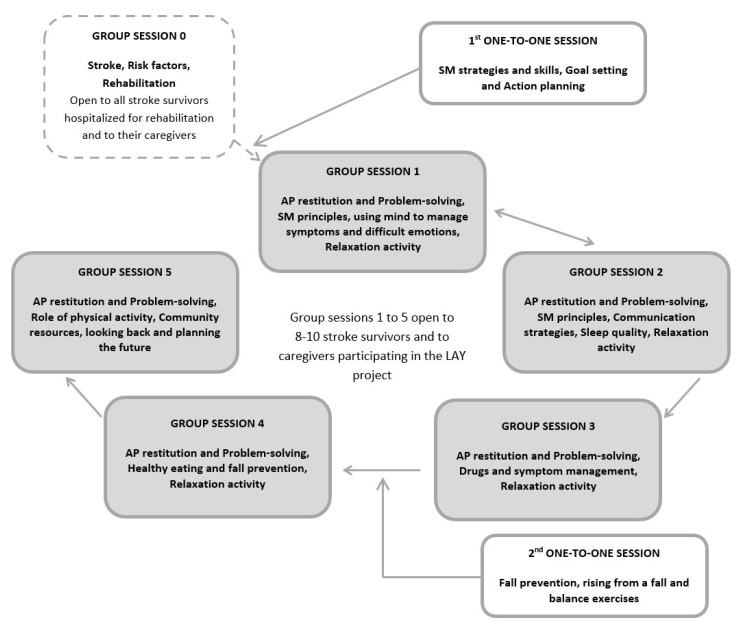
Timetable of the program sessions of the LAY intervention. AP—Action Plan; SM—Self-Management.

**Table 1 ijerph-18-05925-t001:** Characteristics of the randomized controlled trials that informed the LAY intervention.

Study	Sample Size	Experimental Self-Management Program	Duration of the Program	Timing of the Program Initiation	Program Leader	Setting of Delivery	Theoretical Basis/Model
Allen, 2002 [19]	96	Individualized intervention, including an initial home biopsychosocial assessment and education visit, and a team-based development and implementation of an individualized treatment plan focused on health promotion and psychosocial well-being.	3 months	Within 1 month post-discharge from stroke unit	Advanced practice nurse care manager	Participants’ homes	Wagner’s chronic care model
Allen, 2009 [20]	380	As for Allen 2002 [19]. The average time spent on intervention was 8.5 h/patient and included a minimum of 2 home visits and periodic phone calls	6 months	Within 2 months from post-stroke unit admission	Advanced practice nurse care manager	Participants’ homes	Wagner’s chronic care model
Cadilhac, 2011 [21]	143	Weekly 2½-hour group sessions including targeted stroke-specific information and strategies to ensure retention of learning and skills	8 weeks	At least 3 months post stroke	National Stroke Foundation’s Stroke Educator and a trained peer facilitator	Community	Stanford CDSMP
Damush, 2011 [22]	63	6 biweekly 20-min telephone calls guided by a standardized manual and targeted to building self-efficacy using goal setting and behavioral contracting	3 months of intervention + 3 months of telephone monitoring and reinforcement	Within 1 month from stroke	Trained nurse, physician, and social scientist	Telephone calls	Stanford CDSMP
Frank, 2000 [23]	41	Two one-to-one sessions plus weekly telephone calls guided by a workbook including information, coping resources, relaxation techniques, problem-solving skills, and rehearsing planning	1 month	Within 24 months from stroke	Researcher	Participants’ homes	Control cognitions (including self-efficacy)
Harwood, 2011 [24]	139	80-min one-to-one session guided by a specific workbook and designed to engage the patient and his/her family in the process of recovery and self-directed rehabilitation, plus/or 80-min inspirational dvd about stroke, stroke recovery, and promoting self-directed rehabilitation strategies	80 min	6 to 12 weeks post-stroke	Trained research assistants	Community	Self-efficacy principles
Johnston, 2007 [25]	203	3 one-to-one and 2 telephone sessions guided by a workbook. The workbook provided information about stroke and recovery and included activities designed to allow the patient to attain the coping skills to encourage self-management. An audio relaxation cassette tape was provided.	5 weeks	Within 2 weeks from hospital discharge	Researcher	Participants’ homes	Control cognitions (including self-efficacy)
Kendall, 2007 [26]	100	2-h group sessions including both generic chronic condition and stroke-specific self-management education regarding health and well-being, group interaction and support, problem solving	7 weeks	3 months post-stroke	Trained healthcare professionals	Community	Stanford CDSMP
Marsden, 2010 [27]	26	Weekly 2½-hour group session including physical activity and education, always addressing nutritional counseling	7 weeks	At least 1 month post-discharge from all stroke therapy programs	Multidisciplinary stroke team members	Local community public hospital	Not described
McKenna, 2013 [17]	25	Weekly one-to-one sessions up to one hour/week, with the support of a stroke workbook, to promote specific self-management behaviors, such as enabling patients to set personalized goals, plan feasible actions, record progress, and problem solving.	6 weeks	Within 4 weeks of commencing rehabilitation in the community	Trained members of the community stroke team	Community	Self-efficacy principles

Table note: RCT—Randomized controlled trial; etc., etcetera; CDSMP—Chronic Disease Self-Management Program.

**Table 2 ijerph-18-05925-t002:** Summary of the activities and topics addressed during LAY intervention.

Group Sessions				
Session n°	Structure		Specific Topics	Specific Activities
0	Participants’ presentation Specific topic Brainstorming Group discussion		Definition of stroke, its physiopathological mechanisms and risk factors Scope and strategies of rehabilitation	Brainstorming Active participation Group discussion
1	Common Structure	Participants’ presentation Action plans restitution Problem solving Self-management principles Specific topic Relaxation activity	Using mind to manage symptoms Difficult emotions management	
2			Good communication (help request) Enjoying a good quality sleep	
3			Drugs management Pain and fatigue management	
4			Healthy diet Falls prevention and balance exercises	
5			Physical activity Community services	
**Individual sessions**				
Session 1 Before the 1st group session			Program introduction Self-management principles Goal setting Introduction to action plan Support the participant to make his/her own first action plan	
Session 2 Before the 4th group session			Accidental falls prevention Balance exercises	

**Table 3 ijerph-18-05925-t003:** Key elements of the LAY intervention.

Main Sources of Self-Efficacy	Technique/Instrument	In LAY Intervention
Mastery experiences	Breaking the task into smaller, achievable components to achieve a positive result in a task or skill	Weekly realistic action plan
Vicarious experiences	Observe someone perceived to be a peer (model) successfully performing a task, i.e., learning from others’ experiences of the post-stroke recovery period	Interactive group sessions
Verbal persuasion	Persuasion and verification from significant individuals (stroke professional or key family member) to increase an individual’s belief about his/her personal level of skill	Successful action plans shared during group sessions Positive feedback from health professionals
Physiological feedback	Interpretation of individual’s physical sensations and emotions and feelings as positive	Training in positive thinking Training in relaxation techniques
**Self-management abilities**	**Technique/Instrument**	**In LAY Intervention**
Problem solving	Information on stroke, risk factors, care pathway, and consequences of stroke	Repetition of problem-solving technique in individual and group sessions
Decision making	Repetition of how to make decisions	Goal setting and decision making in individual and group sessions
Appropriate resource utilization	Giving information to facilitate knowledge and access to community resources	Oral information in a group session and written information in patients Manual
Partnership with healthcare professionals	Training in how to ask for help	Training patients’ ability to communicate and collaborate in a group session
Taking necessary actions	Action plan as a good instrument to focus on achievable goals	Training in action planning every week for 6 weeks

Table note: LAY—Look After Yourself.

## Data Availability

Data of the LAY clinical trial are contained within the article: Messina, R.; Dallolio, L.; Fugazzaro, S.; Rucci, P.; Iommi, M.; Bardelli, R.; Costi, S.; Denti, M.; Accogli, M.A.; Cavalli, E.; Pagliacci, D.; Fantini, M.P.; Taricco, M.; LAY Project. The Look After Yourself (LAY) intervention to improve self-management in stroke survivors: Results from a quasi-experimental study. Patient Educ Couns. 2020 Jun;103(6):1191–1200. doi:10.1016/j.pec.2020.01.004.

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
