# Peer review of "Self-Management in Stroke Survivors: Development and Implementation of the Look after Yourself (LAY) Intervention"

_ijerph, 2021, doi:10.3390/ijerph18115925_

Round 1

Reviewer 1 Report

This paper describes in detail the components of an intervention for stroke rehabilitation and self-management which should be useful to inform future research and practice in this area.

The first few paragraphs of the Introduction could be combined into 1-2 paragraphs.

Self Management doesn’t require abbreviation, it is only two words and not a common abbreviation. Same with AP

Lines 68-73, please split into more than one sentence

It seems strange to me to have published the main project findings before publishing the present paper, the description of how it was developed which would’ve occurred some time before results were available and subsequently published. Please comment on this timing.

Give more detail on the Chronic Disease Self-Management program and reference source when it is first mentioned in the text.

The literature review focuses on ‘best evidence’. How was evidence quality determined?

Are the studies in table 1 based on any order, e.g. alphabetical? It would be useful to see the two Alen studies together. Please also reference the models in the table. It would also be good to present the main findings from the studies e.g. did they achieve signficant changes or improvements?

Typo errors to amend on lines 155 (drawing not draw) 205 (onfall)

In line 286, who does the ‘whole research group’ refer to? Just researchers or other stakeholders, patients, focus group participants?

A limitation of the program is that it relies on face to face delivery. Could you comment on intervention delivery possibilities when face to face delivery is not possible?

Reviewer 2 Report

Dear authors,

This is one of the most interesting examples of intervention that, in addition to a practical development in the clinic, has a well-founded theoretical framework.

I would like to congratulate the authors. The publication of this type of intervention design can always be of use to other researchers.

I have only one point to make regarding the abstract: a structured abstract would be the best way to reach the reader, with a brief introduction to the objectives of the patients and the interventions of the professionals.
I also suggest changing the word "paper" on line 24 to "report" or another synonym more appropriate for a published paper. 

Author Response

Thank you for your kind reply and for your suggestions.
